analytical chemistry/biochemistry

graphene quantum dot, glucose oxidase, direct electron transfer, biosensor, glucose

**Authors for correspondence:**
Binyuan Xia
e-mail: ybinxia@caep.cn
Shaofei Wang
e-mail: wangshaofei@caep.cn

This article has been edited by the Royal Society of Chemistry, including the commissioning, peer review process and editorial aspects up to the point of acceptance.

# Graphene quantum dot electrochemiluminescence increase by bio-generated H$_2$O$_2$ and its application in direct biosensing

Shanli Yang, Mingfu Chu, Jie Du, Yingru Li, Tao Gai, Xinxin Tan, Binyuan Xia and Shaofei Wang

Institute of Materials, China Academy of Engineering Physics, Jiangyou 621907, People's Republic of China

SY, 0000-0002-9715-6516

In this study, a novel signal-increase electrochemiluminescence (ECL) biosensor has been developed for the detection of glucose based on graphene quantum dot/glucose oxidase (GQD/GO*x*) on Ti foil. The proposed GQD with excellent ECL ability is synthesized through a green one-step strategy by the electrochemical reduction of graphene oxide quantum dot. Upon the addition of glucose, GO*x* can catalytically oxidize glucose and the direct electron transfer between the redox centre of GO*x* and the modified electrode also has been realized, which results in the bio-generated H$_2$O$_2$ for ECL signal increase in GQD and realizes the direct ECL detection of glucose. The signal-increase ECL biosensor enables glucose detection with high sensitivity reaching $5 \times 10^{-6}$ mol l$^{-1}$ in a wide linear range from $5 \times 10^{-6}$ to $1.5 \times 10^{-3}$ mol l$^{-1}$. Additionally, the fabrication process of such GQD-based ECL biosensor is also suitable to other biologically produced H$_2$O$_2$ system, suggesting the possible applications in the sensitive detection of other biologically important targets (e.g. small molecules, protein, DNA and so on).

## 1. Introduction

Graphene quantum dot (GQD), a newly promising zero-dimensional (0D) graphene material, not only shows the similar ability to graphene (e.g. high electron mobility, good chemical inertness and eco-friendly nature) but also possesses many unique merits such as excellent biocompatibility, tuneable bandgap and outstanding photoluminescence/chemiluminescence, owing to its

strong quantum confinement effect and pronounced edge effect [1–4]. Recently, particular interest has been developed in electrochemiluminescence (ECL) ability of GQD due to its promising use in biosensing and bioimaging [5–11].

However, to date, the synthesis of GQD with the ECL property is still at an inchoate stage, not to mention its application in ECL biosensor. Meanwhile, current methods for the ECL GQD production are primarily via scissoring differently huge carbon materials, such as graphene oxide [12,13], XC-72 carbon black [14,15] or coal [16], into small graphene pieces through chemical means [17,18]; nevertheless, these methods often require complex and harsh synthetic procedures, involve the use of toxic organic reagents and, most importantly, generate GQD in large size. Thus, a facile and green approach to synthesize small-sized GQD with the ECL property is still an exigent demand.

Here, we first present a facile one-step strategy for the green synthesis of small-sized ECL GQD based on the electrochemical reduction of graphene oxide quantum dot (GOQD). Moreover, as reported, the ECL intensity of other traditional quantum dots can be linearly enhanced with assistance from $H_2O_2$ [19–22]; moreover, $H_2O_2$ can be biologically produced by various oxidases and their corresponding substrates [23,24]. In this work, glucose oxidase (GOx) has been chosen as a model oxidase to catalyse glucose for the generation of $H_2O_2$, and the direct electron transfer between the redox centre of GOx and the modified electrode also can be realized. More interestingly, the increasing concentration of the bio-generated $H_2O_2$ is well linear with the successive ECL enhancement of GQD, indicating the possible fabrication of an ECL biosensor. To the best of our knowledge, nearly no related work has been reported, and we would like to point out that this is the first report on using the bio-generated $H_2O_2$ for ECL increase in GQD and realizing the direct ECL detection of glucose. In addition, the fabrication process of such GQD-based ECL biosensor is also suitable to other biologically produced $H_2O_2$ system, suggesting the possible applications in the sensitive detection of other biologically important targets (e.g. small molecules, protein, DNA and so on).

# 2. Experimental procedure

## 2.1. Reagents

Titanium (Ti) foil (99.8%, 0.127 mm thickness) and GOx (*Aspergillus niger*, 100 U mg$^{-1}$) were purchased from Aldrich. GOQD was prepared according to the previous work of Zhu *et al*. [25]. D-Glucose was purchased from Shanghai Sangon and dissolved in 0.067 mol l$^{-1}$ pH ~7 phosphate buffer solution (PBS) to form a 1 mol l$^{-1}$ glucose stock solution. All other reagents were of analytical grade and used without further purification. Ultrapure water was used throughout the experiments.

## 2.2. Instruments

Fourier transform infrared spectroscopy (FTIR, FD-5DX), Raman spectroscopy (Labram-010 with a 632.8 nm laser), photoluminescence spectroscopy (PL, Thermo Fisher Scientific Lumina system) and X-ray photoelectron spectroscopy (XPS, Thermo Fisher Scientific K-Alpha 1063 system). Electrochemical measurements were carried out on the three-electrode CHI 660D electrochemistry workstation (Chenhua Instrument Inc., China) using modified Ti foil (0.5 × 1.0 cm) as a working electrode, Pt foil as a counter electrode and saturated calomel electrode (SCE) as a reference electrode. ECL measurements were performed on MPI-E multifunctional chemiluminescent analyser (Xi'an Rimax Electronics Co. Ltd, China).

## 2.3. Preparation of GQD/GOx hybrid

Prior to modification, Ti foil (0.5 × 0.5 cm) was ultrasonically cleaned in acetone and ethanol solution for 15 min, respectively. The cleaned Ti foil was then immersed into the prepared 1 mg ml$^{-1}$ GOQD solution and subjected to cyclic voltammetric scanning from −1.4 to +1.0 V at 50 mV s$^{-1}$ for 10 cycles under stirring. Then, 5 µl of 10 mg ml$^{-1}$ GOx solution was dip-coated onto the modified Ti foil surface using a syringe. After drying at room temperature, the GQD/GOx was obtained. To obtain excellent electrochemical properties, the above experimental conditions were optimized.

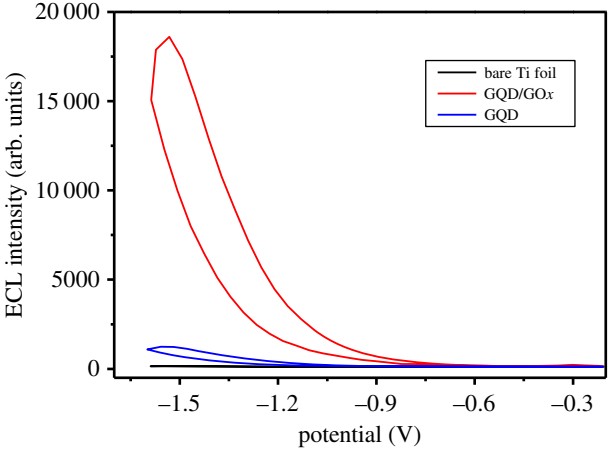

**Figure 1.** ECL–potential curves of bare Ti foil (black line), GQD on Ti foil (blue line) and GQD/GO$x$ on Ti foil (red line) in 0.067 mol l$^{-1}$ pH ~7 PBS with 0.1 mmol l$^{-1}$ H$_2$O$_2$.

# 3. Results and discussion

## 3.1. Characterization and evaluation of GQD and GQD/GO$x$

As reported, GOQD with plenty of oxygenous functional groups on the surface shows excellent water-solubility, while the solubility of GQD was just the opposite [25,26]. Inspired by this, it is logical that the insoluble GQD is likely to be directly prepared on the electrode as long as the soluble GOQD can receive the electron and be electroreduced via the direct contact with the electrode surface. In this work, we have successfully prepared GQD through electroreduction of GOQD, and relatively evincive studies (e.g. cyclic voltammetry, FTIR, Raman, XPS and PL) have been discussed in detail in the electronic supplementary material, figures S1–S5. To the best of our knowledge, this is the first report on using the solubility difference of GOQD and reduced GOQD for the direct preparation of a GQD film.

Electronic supplementary material, figure S6(A) shows the cyclic voltammograms (CVs) of GQD on Ti foil and GQD/GO$x$ on Ti foil in N$_2$-saturated 0.067 mol l$^{-1}$ pH ~7 PBS, respectively. No obvious peak was detected for GQD, while the GQD/GO$x$ showed a pair of stable and quasi-reversible redox peaks, which was due to the direct electron transfer between the redox centre of GO$x$ and the modified electrode; moreover, the cyclic voltammetric current of GQD/GO$x$ was much larger than that of GQD, indicating the higher conductivity and the larger surface-to-volume of GQD/GO$x$. Based on Faraday's Law $\Gamma = Q/(nFA)$ [27], where the surface coverage is $\Gamma$, the charge amount is $Q$, the transferred electron number is $n$, Faraday's constant is $F$ and the effective electrode area is $A$. The $\Gamma$ of electroactive GO$x$ was estimated to be $3.4 \times 10^{-9}$ mol cm$^{-2}$ at GQD/GO$x$-modified Ti foil, which was about 1200-fold larger than the value obtained on the bare electrode surface [28], meaning the excellent biocompatibility and good adsorbability of GQD for GO$x$. Electronic supplementary material, figure S6(B) presents the CVs of GQD/GO$x$ on Ti foil at different scan rates. The redox peak potentials of GO$x$, respectively, shifted in both negative and positive directions; meanwhile, the pair of reversible redox peak currents enhanced successively with the increasing scan rates from 0.05 to 0.5 V s$^{-1}$, suggesting a well reversible and surface-controlled electron transfer process between GO$x$ and the electrode.

## 3.2. ECL behaviours and mechanism

Figure 1 displays the ECL–potential curves of bare Ti foil, GQD on Ti foil and GQD/GO$x$ on Ti foil in 0.067 mol l$^{-1}$ pH ~7 PBS with 0.1 mmol l$^{-1}$ H$_2$O$_2$. Though no ECL signal was detected for bare Ti foil (black line), obvious ECL peaks could be observed on both GQD (blue line) and GQD/GO$x$ (red line), indicating that GQD was an ECL material. Moreover, the ECL intensity of GQD/GO$x$ was about 6.1 times higher than that of GQD, confirming again the higher conductivity and the larger surface-to-volume of GQD/GO$x$. For an ECL system, high ECL emission intensity is essential to achieve a high sensitivity; therefore, high sensitivity can be expected for the GQD/GO$x$ ECL system. Additionally, in the ECL study, the ECL onset potential of GQD/GO$x$ at more positive potential (−0.6 V) was very attractive in comparison with previously reported values based on other heavy metal quantum dot systems, which could result in less interference from other electroactive substances [29–31].

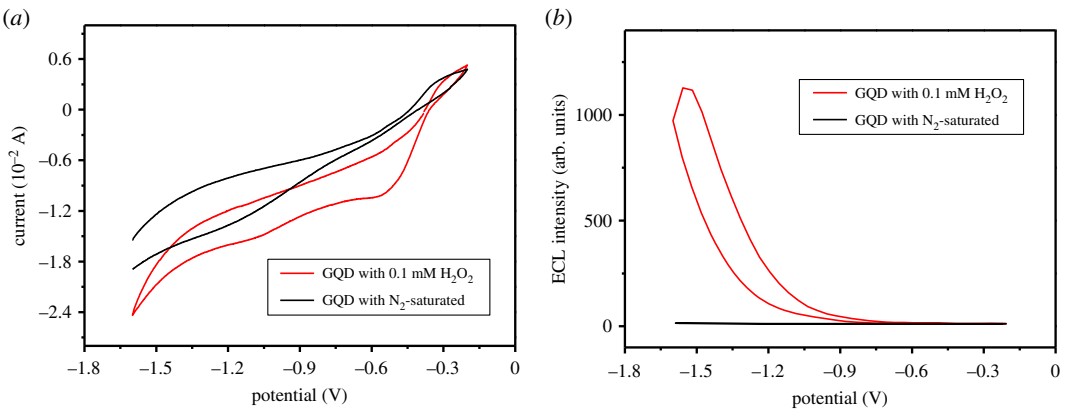

**Figure 2.** (a) CVs and (b) ECL of GQD on Ti foil in 0.067 mol l$^{-1}$ pH $\sim$ 7 PBS with N$_2$-saturated and 0.1 mmol l$^{-1}$ H$_2$O$_2$.

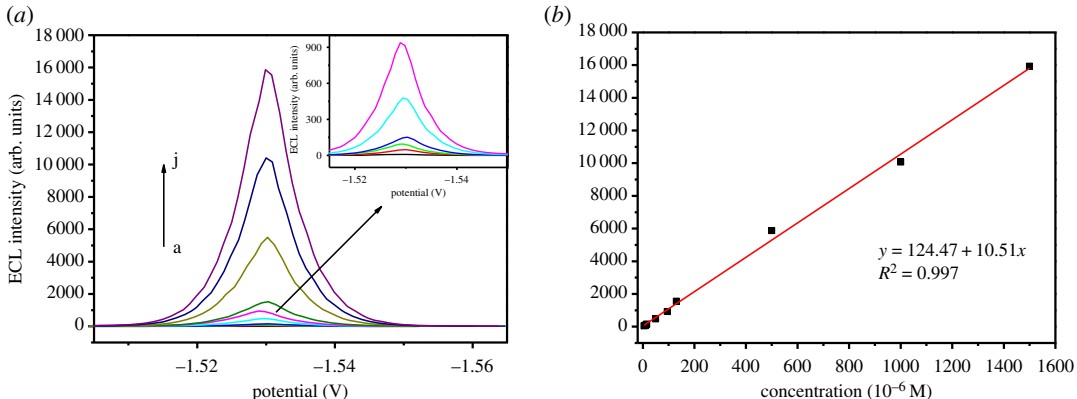

**Figure 3.** (a) ECL–potential curve of GQD/GOx for different concentrations of glucose ($\times 10^{-6}$ mol l$^{-1}$): (A) 5, (B) 10, (C) 15, (D) 50, (E) 50, (F) 100, (G) 150, (H) 500, (I) 1000 and (J) 1500 and (b) the calibration curve for glucose determination.

To learn more about the ECL emission mechanism of GQD, the CVs and ECL of GQD on Ti foil have been examined in 0.067 mol l$^{-1}$ pH ~7 PBS with N$_2$-saturated and 0.1 mmol l$^{-1}$ H$_2$O$_2$, respectively. As shown in figure 2a, an irreversible reduction process at around $-1.05$ V could be observed in N$_2$-saturated solution, which was ascribed to the injection of an electron into GQD to generate reduced state GQD. In 0.1 mmol l$^{-1}$ H$_2$O$_2$ solution, another reduction peak at around $-0.55$ V also could be detected except for the original reduction peak of GQD (approx. $-1.05$ V), owing to the reduction of H$_2$O$_2$. Meanwhile, in figure 2b, due to the non-existence of any coreactant in N$_2$-saturated solution, there was no evident ECL emission during the cathodic ECL process; however, in 0.1 mmol l$^{-1}$ H$_2$O$_2$ solution, the ECL signal appeared beyond $-0.55$ V, increased significantly after $-1.05$ V and achieved the maximum value at around $-1.53$ V, which agreed well with the respective electron injection voltages of H$_2$O$_2$ and GQD in previous CVs. Accordingly, the possible ECL emission mechanisms were shown as below

$$GQD + e^- \rightarrow GQD^{\bullet-}, \tag{3.1}$$
$$H_2O_2 + 2GQD^{\bullet-} \rightarrow GQD^* + 2OH^- \tag{3.2}$$

and
$$GQD^* \rightarrow GQD + h\nu.$$

## 3.3. Detection of glucose

H$_2$O$_2$ can be biologically produced by various oxidases and their corresponding substrates [23,25]. In this work, GOx and glucose have been chosen as a pair of model oxidase and substrate to generate H$_2$O$_2$. Meanwhile, the bio-generated H$_2$O$_2$ concentration increases with the augmentation of glucose concentration, leading to the ECL enhancement of GQD. Thus, a novel ECL biosensor can be fabricated by monitoring the ECL increase in GQD.

Figure 3a shows the ECL–potential curve of GQD/GOx for different concentrations of glucose. During the whole monitoring process, the ECL intensity of GQD/GOx was gradually raised with the

increasing glucose concentration. As shown in figure $3b$, in the range of $5 \times 10^{-6}$–$1.5 \times 10^{-3}$ mol l$^{-1}$, the ECL intensity revealed a linear relationship with the logarithm of glucose concentration with a correlation coefficient of 0.997, and the detection limit (LOD) was $5 \times 10^{-6}$ mol l$^{-1}$ ($S/N = 3$). Reproducibility and stability of this ECL biosensor were also tested. The ECL response of five identical GQD/GO$x$-modified electrodes to $5 \times 10^{-5}$ mol l$^{-1}$ glucose exhibited a relative standard deviation (r.s.d.) of 5.1%, suggesting the acceptable reproducibility of this ECL biosensor. Moreover, to validate the stability of the biosensor under the storage condition (0.067 mol l$^{-1}$ pH ~7 PBS, 4°C), the ECL responses to $5 \times 10^{-5}$ mol l$^{-1}$ glucose were recorded during one month at 2-day intervals. The proposed biosensor could retain about 89% of its original ECL response, resulting from the excellent chemical stability of the GQD/GO$x$ hybrid and the good bioactivity of GO$x$ immobilized on GQD for a long time.

## 4. Conclusion

In this work, a facile one-step strategy for the green synthesis of small-sized ECL GQD is first proposed. The obtained GQD shows good bioactivity to GO$x$, and the direct electron transfer between GO$x$ and the modified electrode surface has been realized. Interestingly, the ECL intensity of GQD is linearly enhanced with assistance from biologically produced H$_2$O$_2$ via a GO$x$ bio-catalysing glucose system; moreover, the bio-generated H$_2$O$_2$ concentration increases with the augmentation of glucose concentration. Thus, a novel ECL biosensor for glucose detection has been fabricated by monitoring the ECL increase in GQD. Additionally, the fabrication of this proposed biosensor also breaks a new path to the sensitive detection of other biologically important targets (e.g. small molecules, protein, DNA and so on) based on such bio-enhanced ECL systems.

Data accessibility. All data are included in the article and the electronic supplementary material.
Authors' contributions. S.Y. and S.W. designed the study and performed the experiments. Y.L. collected and analysed the data. B.X. wrote the paper. M.C. reviewed and edited the manuscript. All authors read and approved the manuscript.
Competing interests. There are no conflicts to declare.
Funding. This work was supported by the National Natural Science Foundation of China (grant nos. 21504085, 11605163, 21604075), Foundation for Special Talents in the China Academy of Engineering Physics (grant nos. TP02201503, TP02201704), the Sichuan Science and Technology Development Foundation for Young Scientists (grant no. 2017JQ0050) and the Development Foundation of Radiochemistry (grant no. XK909) from the China Academy of Engineering Physics.

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
