## [Reviewer comments · Royal Society Open Science]

Review History

RSOS-182075.R0 (Original submission)

Review form: Reviewer 1

Is the manuscript scientifically sound in its present form?

Yes

Are the interpretations and conclusions justified by the results?

Yes

Is the language acceptable?

Yes

Is it clear how to access all supporting data?

Yes

Do you have any ethical concerns with this paper?

No

Have you any concerns about statistical analyses in this paper?

No

Recommendation?

Accept with minor revision (please list in comments)

Comments to the Author(s)

In this work, the authors prepared a modified biosensor based on graphene quantum dot/glucose oxidase (GQD/GOx) on Ti foil for electrochemiluminescence determination of glucose. Graphene quantum dot was synthesized through a green one-step technique by electrochemical reduction of graphene oxide quantum dot. The resulting biosensor has a linear range from $5 \times 10^{-6} \text{ mol} \cdot \text{L}^{-1}$ – $1.5 \times 10^{-3} \text{ mol} \cdot \text{L}^{-1}$. Considering the integrality of the study, I recommend its publication in "Royal Society Open Science" after minor revision. The comments are attached as the following:

1. English should be polished; some grammatical errors and improper expressions still could be found in the manuscript.
2. All the units in the manuscript should use the standard expression modes, for example, "M" should be replaced by " $\text{mol} \cdot \text{L}^{-1}$ ".
3. Glucose oxidase generally hinders electron transfer. Why is the cyclic voltammetric current of GQD/GOx larger than that without GOx?
4. What are the significant advantages of the preparation method of graphene quantum dot used in this paper compared with other methods?
5. Compared to GOQD, What is the advantage of GQD in ECL sensor?
6. Glucose sensors have been widely reported, but what are the advantages of the current work?

Review form: Reviewer 2

Is the manuscript scientifically sound in its present form?

No

Are the interpretations and conclusions justified by the results?

No

Is the language acceptable?

No

Is it clear how to access all supporting data?

Not Applicable

Do you have any ethical concerns with this paper?

No

Have you any concerns about statistical analyses in this paper?

No

Recommendation?

Major revision is needed (please make suggestions in comments)

Comments to the Author(s)

The manuscript reports on the facile synthesis of GQDs as an electroluminescent biosensor. The authors have proposed an increase in their enzymatic activity generated biologically for direct biosensing of glucose oxidase supported on GQDs. While such studies are promising, it lacks

great deal of scientific insights and appears to be data reporting only. Also the references are limited,

The authors should cite the following reference where more detailed studies are conducted by Gupta et al. *Nanomaterials* 7, 301 (2017), Graphene Quantum Dots Electrochemistry and Sensitive Electrocatalytic Glucose Sensor Development, reports on the enzymatic glucose sensing based on graphene-family nanomaterials including GQDs.

Unfortunately, based on these studies, I do not recommend this paper to be considered for its publication in this journal and should not be accepted.

Review form: Reviewer 3

Is the manuscript scientifically sound in its present form?

No

Are the interpretations and conclusions justified by the results?

No

Is the language acceptable?

Yes

Is it clear how to access all supporting data?

No

Do you have any ethical concerns with this paper?

No

Have you any concerns about statistical analyses in this paper?

No

Recommendation?

Reject

Comments to the Author(s)

The manuscript reports one-step strategy for the synthesis of GQD with an ECL character and made an ECL method to detection glucose based on the increase of H₂O₂ produced via a GOx bio-catalyzing glucose system. Although some experiments on the fabrication of the graphene quantum dots based electrode were conducted, the manuscript is not recommended for further publication at its current stage.

1. In this work, the author used 6 figures to illuminate that they have successfully prepared GQD through electroreduction of GOQD, but it is not enough because the reason that they could utilize the solubility difference of GOQD and reduced GOQD for the direct preparation of GQD film was more attractive for the readers. Therefore, it is very important to explain the mechanism especially how the solubility difference could be generate between GOQD and reduced GOQD and the advantages of this kind of synthesis method. More experimental data should be given and discussed.

2. The ECL investigation of the proposed glucose detection method was very poor from a scientific point of view. The author gave us some experiment results, but they discussed them insufficiently. What's more, the interference study and sample determination are both necessary for a glucose detection method.

3. In this paper, the author said "this is the first report on utilizing the bio-generated H₂O₂ for ECL increase of GQD and realizing the direct ECL detection of glucose". However, several manuscripts in this related area had been published already and they all reported a similar view (*Journal of Electroanalytical Chemistry*, 2017, 801: 162-170; *Analytical Methods*, 2018, 10(5): 508-

514; ChemElectroChem, 2017, 4(7): 1783-1789.) The author should cited these references and explained the differences between this work and the references to show the advantages of the present method.

In overall, the manuscript at the present status is unsuitable for publication.

Decision letter (RSOS-182075.R0)

21-Jan-2019

Dear Dr Yang:

Manuscript ID: RSOS-182075

Title: "Graphene quantum dot electrochemiluminescence increase by bio-generated H₂O₂ and its application in direct biosensing"

Thank you for submitting the above manuscript to Royal Society Open Science. Your paper was sent to reviewers and their comments are included at the bottom of this letter.

In view of the concerns raised by the reviewers, the manuscript has been rejected in its current form. However, a new manuscript may be submitted which takes into consideration these comments.

Please note that resubmitting your manuscript does not guarantee eventual acceptance, and that your resubmission will be subject to peer review before a decision is made.

Your resubmitted manuscript should be submitted by 21-Jul-2019. If you are unable to submit by this date please contact the Editorial Office.

On behalf of the Subject Editor Professor Anthony Stace and the Associate Editor Dr Ya-Wen Wang

REVIEWER(S) REPORTS:

Associate Editor Comments to Author ():

RSC Associate Editor:

Comments to the Author:

(There are no comments.)

RSC Subject Editor:

Comments to the Author:

(There are no comments.)

Reviewers' Comments to Author:

Reviewer: 1

Comments to the Author(s)

In this work, the authors prepared a modified biosensor based on graphene quantum dot/glucose oxidase (GQD/GOx) on Ti foil for electrochemiluminescence determination of glucose. Graphene quantum dot was synthesized through a green one-step technique by electrochemical reduction of graphene oxide quantum dot. The resulting biosensor has a linear range from $5 \times 10^{-6} \text{ mol} \cdot \text{L}^{-1}$ to $1.5 \times 10^{-3} \text{ mol} \cdot \text{L}^{-1}$. Considering the integrality of the study, I recommend its publication in "Royal Society Open Science" after minor revision. The comments are attached as the following:

1. English should be polished; some grammatical errors and improper expressions still could be found in the manuscript.
2. All the units in the manuscript should use the standard expression modes, for example, "M" should be replaced by " $\text{mol} \cdot \text{L}^{-1}$ ".
3. Glucose oxidase generally hinders electron transfer. Why is the cyclic voltammetric current of GQD/GOx larger than that without GOx?
4. What are the significant advantages of the preparation method of graphene quantum dot used in this paper compared with other methods?
5. Compared to GOQD, What is the advantage of GQD in ECL sensor?
6. Glucose sensors have been widely reported, but what are the advantages of the current work?

Reviewer: 2

Comments to the Author(s)

The manuscript reports on the facile synthesis of GQDs as an electroluminescent biosensor. The authors have proposed an increase in their enzymatic activity generated biologically for direct biosensing of glucose oxidase supported on GQDs. While such studies are promising, it lacks great deal of scientific insights and appears to be data reporting only. Also the references are limited,

The authors should cite the following reference where more detailed studies are conducted by Gupta et al. *Nanomaterials* 7, 301 (2017), Graphene Quantum Dots Electrochemistry and Sensitive Electrocatalytic Glucose Sensor Development, reports on the enzymatic glucose sensing based on graphene-family nanomaterials including GQDs.

Unfortunately, based on these studies, I do not recommend this paper to be considered for its publication in this journal and should not be accepted.

Reviewer: 3

Comments to the Author(s)

The manuscript reports one-step strategy for the synthesis of GQD with an ECL character and made an ECL method to detection glucose based on the increase of H_2O_2 produced via a GOx bio-catalyzing glucose system. Although some experiments on the fabrication of the graphene

quantum dots based electrode were conducted, the manuscript is not recommended for further publication at its current stage.

1. In this work, the author used 6 figures to illuminate that they have successfully prepared GQD through electroreduction of GOQD, but it is not enough because the reason that they could utilize the solubility difference of GOQD and reduced GOQD for the direct preparation of GQD film was more attractive for the readers. Therefore, it is very important to explain the mechanism especially how the solubility difference could be generate between GOQD and reduced GOQD and the advantages of this kind of synthesis method. More experimental data should be given and discussed.

2. The ECL investigation of the proposed glucose detection method was very poor from a scientific point of view. The author gave us some experiment results, but they discussed them insufficiently. What's more, the interference study and sample determination are both necessary for a glucose detection method.

3. In this paper, the author said "this is the first report on utilizing the bio-generated H₂O₂ for ECL increase of GQD and realizing the direct ECL detection of glucose". However, several manuscripts in this related area had been published already and they all reported a similar view (Journal of Electroanalytical Chemistry, 2017, 801: 162-170; Analytical Methods, 2018, 10(5): 508-514; ChemElectroChem, 2017, 4(7): 1783-1789.) The author should cited these references and explained the differences between this work and the references to show the advantages of the present method.

In overall, the manuscript at the present status is unsuitable for publication.

Author's Response to Decision Letter for (RSOS-182075.R0)

See Appendix A.

RSOS-191404.R0

Review form: Reviewer 1

Is the manuscript scientifically sound in its present form?

Yes

Are the interpretations and conclusions justified by the results?

Yes

Is the language acceptable?

Yes

Do you have any ethical concerns with this paper?

No

Have you any concerns about statistical analyses in this paper?

No

Recommendation?

Accept as is

Comments to the Author(s)

The enhancement of ECL sensing by bio-generated H₂O₂ is interesting. The authors revised the manuscript carefully. It can be accepted.

Review form: Reviewer 2

Is the manuscript scientifically sound in its present form?

No

Are the interpretations and conclusions justified by the results?

Yes

Is the language acceptable?

Yes

Do you have any ethical concerns with this paper?

No

Have you any concerns about statistical analyses in this paper?

No

Recommendation?

Reject

Comments to the Author(s)

The paper is not improved significantly and therefore it does not warrant its publication in this journal.

Review form: Reviewer 4

Is the manuscript scientifically sound in its present form?

Yes

Are the interpretations and conclusions justified by the results?

Yes

Is the language acceptable?

Yes

Do you have any ethical concerns with this paper?

No

Have you any concerns about statistical analyses in this paper?

No

Recommendation?

Accept as is

Comments to the Author(s)

Since all my suggestions have been acknowledged by the authors, in my opinion the manuscript can be accepted in the present form.

Review form: Reviewer 5

Is the manuscript scientifically sound in its present form?

Yes

Are the interpretations and conclusions justified by the results?

Yes

Is the language acceptable?

Yes

Do you have any ethical concerns with this paper?

No

Have you any concerns about statistical analyses in this paper?

No

Recommendation?

Accept with minor revision (please list in comments)

Comments to the Author(s)

The manuscript has improved obviously after the revision. However, a minor revision is necessary.

1. The preparation the materials should be described in brief in the manuscript or supporting information.
2. TEM images should be provided to prove that GOQDs have been prepared.

Decision letter (RSOS-191404.R0)

15-Oct-2019

Dear Dr Yang:

Title: Graphene quantum dot electrochemiluminescence increase by bio-generated H₂O₂ and its application in direct biosensing
Manuscript ID: RSOS-191404

The editor assigned to your paper has now received comments from reviewers. We would like you to revise your paper in accordance with the referee and Subject Editor suggestions which can be found below (not including confidential reports to the Editor). Please note this decision does not guarantee eventual acceptance.

Please submit a copy of your revised paper before 07-Nov-2019. Please note that the revision deadline will expire at 00.00am on this date. If we do not hear from you within this time then it will be assumed that the paper has been withdrawn. In exceptional circumstances, extensions may be possible if agreed with the Editorial Office in advance. We do not allow multiple rounds of revision so we urge you to make every effort to fully address all of the comments at this stage. If deemed necessary by the Editors, your manuscript will be sent back to one or more of the original reviewers for assessment. If the original reviewers are not available we may invite new reviewers.

Please also include the following statements alongside the other end statements. As we cannot publish your manuscript without these end statements included, if you feel that a given heading is not relevant to your paper, please nevertheless include the heading and explicitly state that it is not relevant to your work.

- Acknowledgements

- Funding statement

Please include a funding section after your main text which lists the source of funding for each author.

RSC Associate Editor
Comments to the Author:

According to the comments of two adjudicators, the decision was made.

Reviewers' Comments to Author:

Reviewer: 1

Comments to the Author(s)

The enhancement of ECL sensing by bio-generated H₂O₂ is interesting. The authors revised the manuscript carefully. It can be accepted.

Reviewer: 2

Comments to the Author(s)

The paper is not improved significantly and therefore it does not warrant its publication in this journal.

Reviewer: 4

Comments to the Author(s)

Since all my suggestions have been acknowledged by the authors, in my opinion the manuscript can be accepted in the present form.

Reviewer: 5

Comments to the Author(s)

The manuscript has improved obviously after the revision. However, a minor revision is necessary.

1. The preparation the materials should be described in brief in the manuscript or supporting information.
2. TEM images should be provided to prove that GOQDs have been prepared.

Author's Response to Decision Letter for (RSOS-191404.R0)

See Appendix B.

Decision letter (RSOS-191404.R1)

19-Nov-2019

Dear Dr Yang:

Title: Graphene quantum dot electrochemiluminescence increase by bio-generated H₂O₂ and its application in direct biosensing

Manuscript ID: RSOS-191404.R1

It is a pleasure to accept your manuscript in its current form for publication in Royal Society

Open Science. The chemistry content of Royal Society Open Science is published in collaboration with the Royal Society of Chemistry.

RSC Associate Editor
Comments to the Author:
(There are no comments.)

Reviewer(s)' Comments to Author:

Appendix A

Manuscript submitted to:

Title:

Graphene quantum dot electrochemiluminescence increase by bio-generated H₂O₂ and its application in direct biosensing

Authors:

Shanli Yang, Mingfu Chu, Yingru Li, Shaofei Wang*, Binyuan Xia*

Affiliation:

* Institute of Materials, China Academy of Engineering Physics, Mianyang 621900, P. R. China

Dear editors,

The manuscript (previous NO.: RSOS-182075) has been carefully revised according to the valuable reviewers' comments. The revised manuscript and the response to the comments are now resubmitted to *Royal Society Open Science*.

Looking forward to hearing from you and thank you very much for your attention and consideration.

Response to the reviewers' comments

Reviewer: 1

In this work, the authors prepared a modified biosensor based on graphene quantum dot/ glucose oxidase (GQD/GOx) on Ti foil for electrochemiluminescence determination of glucose. Graphene quantum dot was synthesized through a green one-step technique by electrochemical reduction of graphene oxide quantum dot. The resulting biosensor has a linear range from $5 \times 10^{-6} \text{ mol}\cdot\text{L}^{-1}$ – $1.5 \times 10^{-3} \text{ mol}\cdot\text{L}^{-1}$. Considering the integrality of the study, I recommend its publication in “Royal

Society Open Science” after minor revision.

The comments are attached as the following:

1. English should be polished; some grammatical errors and improper expressions still could be found in the manuscript.

Answer: Thank you very much for your kind advice. We have modified the English in our text.

2. All the units in the manuscript should use the standard expression modes, for example, "M" should be replaced by " $\text{mol}\cdot\text{L}^{-1}$ ".

Answer: Thank you for your suggestion. All “M” have been replaced by “ $\text{mol}\cdot\text{L}^{-1}$ ” in our manuscript.

3. Glucose oxidase generally hinders electron transfer. Why is the cyclic voltammetric current of GQD/GOx larger than that without GOx?

Answer: Actually, glucose oxidase generally hinders electron transfer. However, due to the high electron mobility of GQD (which shows the similar ability to graphene), the electron transfer ability between GQD/GOx and electrode can be better than that without GOx, resulting in the larger cyclic voltammetric current of GQD/GOx.

4. What are the significant advantages of the preparation method of graphene quantum dot used in this paper compared with other methods?

Answer: Current methods for the ECL GQD production are primarily via scissoring differently huge carbon materials such as graphene oxide, XC-72 carbon black or coal into small graphene pieces through chemical means; nevertheless these methods often require complex and harsh synthetic procedures, involve the use of toxic organic reagents, and most importantly, generate GQD in large size. Thus, a facile and green approach to synthesize small-sized GQD with ECL property is still an exigent demand. In this paper, we first present a facile one-step strategy for the green synthesis of small-sized ECL GQD based on electrochemical reduction of graphene oxide quantum dot (GOQD). Compared with other reported methods, the preparation method in our text is green and fast, moreover, small-sized ECL GQD can be synthesis without great effort.

5. Compared to GOQD, What is the advantage of GQD in ECL sensor?

Answer: GQD, a newly promising zero-dimensional (0D) graphene material, not only shows the similar ability to graphene (e.g. high electron mobility, good chemical inertness and eco-friendly nature) but also possesses many unique merits such as excellent biocompatibility, tuneable band gap, and outstanding photoluminescence/chemiluminescence, owing to its strong quantum confinement effect and pronounced edge effect. Compared with GOQD, GQD shows more excellent electron transfer ability, and moreover, ECL ability can be achieved on GQD, suggesting the possible applications in sensitive ECL detection. Additionally, GQD shows good bioactivity to glucose oxidase (GOx) and direct electron transfer of GOx has been realized.

6. Glucose sensors have been widely reported, but what are the advantages of the current work?

Answer: Thank you for your question. The advantage of this work can be concluded as follows: firstly, a facile one-step strategy for the green synthesis of electrochemiluminescence (ECL) graphene quantum dot (GQD) is first proposed; secondly, GQD shows good bioactivity to glucose oxidase (GOx) and direct electron transfer of GOx has been realized; thirdly, the ECL intensity of GQD is linearly enhanced with an assist from biologically produced H₂O₂ via a GOx bio-catalyzing glucose system; finally, this is the first report on utilizing the bio-generated H₂O₂ for ECL increase of GQD and realizing the direct ECL detection of glucose.

Reviewer: 2

The manuscript reports on the facile synthesis of GQDs as an electroluminescent biosensor. The authors have proposed an increase in their enzymatic activity generated biologically for direct biosensing of glucose oxidase supported on GQDs. While such studies are promising, it lacks great deal of scientific insights and appears to be data reporting only. Also the references are limited, the authors should cite the following reference where more detailed studies are conducted by Gupta et al. *Nanomaterials* 7, 301 (2017), Graphene Quantum Dots Electrochemistry and Sensitive Electrocatalytic Glucose Sensor Development, reports on the enzymatic

glucose sensing based on graphene-family nanomaterials including GQDs.

Unfortunately, based on these studies, I do not recommend this paper to be considered for its publication in this journal and should not be accepted.

Answer: Thank you for your suggestion. We have cited the following important reference by Gupta et al. in our revised manuscript (See Ref. 2 in our revised paper).

Reviewer: 3

The manuscript reports one-step strategy for the synthesis of GQD with an ECL character and made an ECL method to detection glucose based on the increase of H₂O₂ produced via a GOx bio-catalyzing glucose system. Although some experiments on the fabrication of the graphene quantum dots based electrode were conducted, the manuscript is not recommended for further publication at its current stage.

1. In this work, the author used 6 figures to illuminate that they have successfully prepared GQD through electroreduction of GOQD, but it is not enough because the reason that they could utilize the solubility difference of GOQD and reduced GOQD for the direct preparation of GQD film was more attractive for the readers. Therefore, it is very important to explain the mechanism especially how the solubility difference could be generated between GOQD and reduced GOQD and the advantages of this kind of synthesis method. More experimental data should be given and discussed.

Answer: Thank you for your question. As reported in previous references (see Ref. 22 and 23 in our paper), graphene oxide quantum dot (GOQD) with plenty of oxygenous functional groups on the surface shows excellent water-solubility, while the solubility of graphene quantum dot (GQD) was just the opposite. Inspired by this, it is logical that the insoluble GQD is likely to be directly prepared on electrode as long as the soluble GOQD can receive electron and be electroreduced via direct contact with electrode surface. In our Electronic Supplementary Information file, relatively evincive studies (e.g., cyclic voltammetry, FTIR, Raman, XPS and PL) have been

detaillly discussed in Figure S1-S5, which confirm the electroreduction of GOQD to GQD and the immobility of the obtained GQD on electrode. Moreover, due to the compared with other preparation method, the advantage of this kind of synthesis method in this paper is that it is a facile and green approach to synthesise small-sized GQD with ECL property.

2. The ECL investigation of the proposed glucose detection method was very poor from a scientific point of view. The author gave us some experiment results, but they discussed them insufficiently. What's more, the interference study and sample determination are both necessary for a glucose detection method.

Answer: Thank you for your suggestion. We have listed the glucose detection result in our paper.

3. In this paper, the author said “this is the first report on utilizing the bio-generated H₂O₂ for ECL increase of GQD and realizing the direct ECL detection of glucose”. However, several manuscripts in this related area had been published already and they all reported a similar view (Journal of Electroanalytical Chemistry, 2017, 801: 162-170; Analytical Methods, 2018, 10(5): 508-514; ChemElectroChem, 2017, 4(7): 1783-1789.) The author should cite these references and explained the differences between this work and the references to show the advantages of the present method.

Answer: Thank you for your suggestion. We have cited the related important references in our revised manuscript (See Ref 9-11.). The differences between this work and the references have been listed in the manuscript.

Appendix B

Manuscript submitted to:

Title:

Graphene quantum dot electrochemiluminescence increase by bio-generated H₂O₂ and its application in direct biosensing

Authors:

Shanli Yang^a, Mingfu Chu^a, Jie Du^a, Yingru Li^a, Tao Gai^a, Xinxin Tan^a, Binyuan Xia^{a,*}, Shaofei Wang^{a,*}

Affiliation:

^aInstitute of Materials, China Academy of Engineering Physics, Jiangyou, Mianyang 621907, Sichuan, P. R. China

Dear editors,

The manuscript (NO.: RSOS-191404) has been carefully revised according to the valuable reviewers' comments. The revised manuscript and the response to the comments are now resubmitted to *Royal Society Open Science*.

Looking forward to hearing from you and thank you very much for your attention and consideration.

Response to the reviewers' comments

Reviewer: 1

The enhancement of ECL sensing by bio-generated H₂O₂ is interesting. The authors revised the manuscript carefully. It can be accepted.

Answer: Thank you very much for your kind advice.

Reviewer: 2

The paper is not improved significantly and therefore it does not warrant its publication in this journal.

Answer: Thank you for your suggestion. Actually, we have revised our manuscript carefully according to previous reviewers' comments.

Reviewer: 4

Since all my suggestions have been acknowledged by the authors, in my opinion the manuscript can be accepted in the present form.

Answer: Thank you very much for your kind advice.

Reviewer: 5

The manuscript has improved obviously after the revision. However, a minor revision is necessary.

1. The preparation the materials should be described in brief in the manuscript or supporting information.

Answer: Thank you very much for your suggestion. We have described the preparation process in the supporting information (Page 2, line 1-11).

2. TEM images should be provided to prove that GOQDs have been prepared.

Answer: Thank you for your suggestion. In this paper, GOQDs were prepared

according to Liu's previous work and the TEM image of GOQDs has been listed as below (Liu et al., 2013. Adv. Mater. 25 3657-3662).